# Complement 3 and the Prognostic Nutritional Index Distinguish Kawasaki Disease from Other Fever Illness with a Nomogram

**DOI:** 10.3390/children8090825

**Published:** 2021-09-20

**Authors:** Yi-Shuang Huang, Xiao-Ping Liu, Han-Bing Xia, Li-Na Cui, Xin-Ling Lang, Chun-Yi Liu, Wei-Dong Huang, Jia-Yu Zhang, Xi Liu, Ho-Chang Kuo, Tao Zhou

**Affiliations:** 1The Department of Emergency and Pediatrics, Shenzhen Baoan Women’s and Children’s Hospital, Jinan University, Shenzhen 518102, China; yishuanghuang@sina.com (Y.-S.H.); yaoguqian@aliyun.com (X.-P.L.); xiahanbingbaoan@sina.com (H.-B.X.); linacuibaoan@sina.com (L.-N.C.); langxinlingbaoan@sina.com (X.-L.L.); wustlxm@aliyun.com (C.-Y.L.); astarry@aliyun.com (W.-D.H.); jiayuzhangbaoan@sina.com (J.-Y.Z.); liuxiaomin@aliyun.com (X.L.); 2Kawasaki Disease Center, Department of Pediatrics, Kaohsiung Chang Gung Memorial Hospital, Kaohsiung 83301, Taiwan; 3College of Medicine, Chang Gung University, Taoyuan 33302, Taiwan

**Keywords:** Kawasaki disease, nomogram model, white blood cell, prognostic nutritional index, platelet-to-lymphocyte ratio, neutrophil-to-lymphocyte ratio, complement 3

## Abstract

Objective: This study aimed to establish a model to distinguish Kawasaki disease (KD) from other fever illness using the prognostic nutritional index (PNI) and immunological factors. Method: We enrolled a total of 692 patients (including 198 with KD and 494 children with febrile diseases). Of those, 415 patients were selected to be the training group and 277 patients to be the validation group. Laboratory data, including the neutrophil-to-lymphocyte ratio (NLR), the platelet-to-lymphocyte ratio (PLR), the prognostic nutritional index (PNI), and immunological factors, were retrospectively collected for an analysis after admission. We used univariate and multivariate logistic regressions and nomograms for the analysis. Result: Patients with KD showed significantly higher C3 and a lower PNI. After a multivariate logistic regression, the total leukocyte count, PNI, C3, and NLR showed a significance (*p* < 0.05) and then performed well with the nomogram model. The areas under the ROC in the training group and the validation group were 0.858 and 0.825, respectively. The calibration curves of the two groups for the probability of KD showed a near agreement to the actual probability. Conclusions: Compared with children with febrile diseases, patients with KD showed increased C3 and a decreased nutritional index of the PNI. The nomogram established with these factors could effectively identify KD from febrile illness in children.

## 1. Introduction

Kawasaki disease (KD), also known as mucocutaneous lymph node syndrome (MCLS), is a form of acute febrile and systemic vasculitis that commonly occurs in children under 5 years old [1]. The primary pathological changes of KD are systemic non-specific vasculitis involving small and medium arteries and the most serious complication is a coronary artery lesion (CAL) including artery aneurysms, coronary artery stenosis, thrombosis, myocardial infarction, and sudden death [2]. With an increasing number of patients in recent years, KD has become the main cause of acquired heart disease in children [3].

Both the etiology and pathogenesis of KD remain unknown but may be the result of combined effects from genetic heredity, infection (bacteria, mycoplasma, virus, COVID-19, fungus, etc.), and the immune response [4]. It is believed that certain pathogenic microbial infections and an immune response imbalance may lead to KD [5,6]. Therefore, speculation has indicated that inflammatory markers may have the potential to identify KD and the disease prognosis [7]. Recent studies have shown that the peripheral blood neutrophil-to-lymphocyte ratio (NLR) and the platelet-to-lymphocyte ratio (PLR) were associated with the severity of KD and the involvement of CALs [8].

The serum level of albumin is a negative acute-phase protein marker that decreases during inflammation or malnutrition. Inflammation reduces the albumin concentration by lowering its rate of synthesis, causing leakage due to the increased permeability of the blood vessels, and is associated with higher catabolic rates [9]. The serum levels of albumin play an important role in diagnosing incomplete KD according to the American Heart Association (AHA) supplementary criteria and predicting IVIG-resistant KD [10,11]. Onodera′s prognostic nutritional index (Onodera′s PNI) is composed of a serum albumin value (ALB) and a peripheral blood lymphocyte count (TLC) with the formula PNI = ALB (g/L) + 5 × TLC (10^9^/L). This index was first proposed by Buzby and later established by Onodera [12,13,14]. The PNI is a scoring index used to assess the nutritional status of patients and predict the risk of surgery and the prognosis of a variety of malignancies [15]. The PNI has been found to be a potentially important predictor of disease activities and complications of autoimmune diseases [16]. Tai et al. reported that the PNI could be a candidate as an adjunctive predictor of a CAL as well as IVIG resistance. Together with a low PNI, factors such as intravenous immunoglobulin (IVIG) resistance, male gender, and platelet count contribute to high odds for predicting a CAL within six months of illness [17]. 

A KD diagnosis primarily depends on the clinical presentation and the exclusion of other clinically similar cases with known causes. The prompt administration of IVIG treatment can reduce the incidence of coronary artery aneurysms from 20–25% to 3–5% [10]. The aim of this study is to explore the clinical value of the PNI combined with immune factors in the identification of KD.

## 2. Materials and Methods

### 2.1. Study Participants

KD children hospitalized in Shenzhen Baoan Women’s and Children’s Hospital from August 2016 to July 2019 were enrolled in this study. Febrile children who were hospitalized on the same day were also enrolled as a control group. Patients with autoimmune diseases, sepsis, or incomplete data were excluded for a total of 51 KD patients and 105 fever controls. The clinical indicators were collected and compared between the two groups. The clinical diagnosis of KD was based on the revised diagnostic criteria of KD by the American Heart Association (AHA) in 2017. IVIG responsiveness was defined as the abatement of fever within 48 h after completing IVIG treatment and no return of fever (> 38 °C) for at least 7 days after with a marked improvement or normalization of the comorbid signs of inflammation. Incomplete KD was defined as those who had less than four symptoms and were finally diagnosed as KD. The clinical data of gender, age, weight, and clinical manifestations of the enrolled children were collected for the analysis. We also recorded the laboratory results including white blood cell (WBC), neutrophil, lymphocyte, platelet, albumin (ALB), immunoglobulin A (IgA), IgG, IgM, C3, C4, the neutrophil-to-lymphocyte ratio (NLR), the platelet-to-lymphocyte ratio (PLR), and the PNI for the analysis. The training group was selected 60% randomly from the entire sample studied.

### 2.2. Statistical Analysis

We used The statistical analyses and graphics were performed with IBM SPSS 13.0 (SPSS Inc, Armonk, NY) and R 3.5.1 (The R Foundation for Statistical Computing, Vienna, Austria) with the rms statistical packages. for the statistical analysis including a single factor analysis and a multifactor analysis, and expressed normal distribution measurement data as mean and standard deviation. An independent sample *t*-test was used to compare the two groups. The median and interquartile range (IQR) were used to describe the measurement data, and the non-parametric Mann–Whitney *U* rank sum test was used for the comparison between groups without a normal distribution. N and the percentage were used to describe the counting data, and a Pearson chi-squared test (χ^2^) was used to compare the counting data. We adopted R 3.5.1 software to make the nomograms, calibration curves, and ROC. A value of *p* < 0.05 was considered statistically significant.

## 3. Results

### 3.1. Clinical Features

We included a total of 692 children in this study including 422 male children (60.98%) and 270 female children (39.02%). The KD group consisted of 118 boys (59.6%) and 98 girls (40.4%) with a median age of 21 (13–17) months. The febrile control group had 304 boys (61.54%) and 190 girls (38.46%) with a median age of 21 (11–41) months. Among them, 415 (60%) children were randomly selected as the training group and 295 (40%) as the verification group. The comparison of variables between the groups is shown in Table 1. No significant difference was observed between the training group and the verification group (*p* > 0.05).

### 3.2. Univariate Analysis

A single variable analysis of all variables in the training group revealed that gender, age, body weight, IgA, IgG, IgM, and C4 were not statistically significant between the KD group and the control group (*p* > 0.05). WBC, the PNI, NLR, PLR, and C3 demonstrated a significant difference between the KD patients and the febrile controls (*p* < 0.05) (Table 2).

### 3.3. Multivariate Logistic Regression Analysis

Significant indicators of the univariate analysis including WBC, C3, NLR, PLR, and the PNI were included in the multivariate analysis to screen out the independent risk factors of KD and a nomogram was established based on the results of the multivariate analysis. The statistics showed that WBC, C3, NLR, and the PNI were independent risk factors of KD (Table 3).

### 3.4. Scoring System for Predicting KD

The logistic regression results of the training group were used to make the nomogram (Figure 1). In the model, the maximum scores corresponding with each predictor were WBC (100 points), C3 (51 points), NLR (60 points), and the PNI (75 points). The occurrence probability of KD corresponding with the scores is shown in Table 4.

### 3.5. Performance of the Nomogram

The ROC curve was applied to verify the nomogram results in the training group and the validation group and showed a good differentiation with an area under the curve of the ROC of 0.815 (95% confidence interval: 0.815–0.901) in the training group and 0.825 (95% confidence interval: 0.769–0.881) in the validation group (Figure 2). The nomogram was calibrated using the calibration curve [18]. A calibration curve of the nomogram for the training group and the validation group is presented in Figure 3, which shows that the prediction of KD by the nomogram agreed well with the actual probabilities in both groups. The calibration curves for the KD outcome in the two groups demonstrated almost no apparent departure from the fit with a good correspondence between the predicted outcome and the actual outcome.

### 3.6. ROC Curve and the Cutoff Value of C3 and the PNI

The area under the curve of the ROC of C3 was 0.761 (95% confidence interval: 0.72–0.801), 0.63 (95% confidence interval: 0.584–0.677) in the PNI, and 0.788 (95% confidence interval: 0.749–0.826) in the two variables combined (Figure 4). The cutoff value of C3 and the PNI was 1.38g/L and 52.03, respectively (Table 5).

## 4. Discussion

The accurate diagnosis of KD remains a challenge for clinicians because its clinical manifestations are often similar to or overlap with other febrile infectious diseases in children and no specific laboratory test is currently available to confirm the diagnosis. A recent study reported the existence of the overdiagnosis of KD [19]. The challenge for clinicians is to prevent the occurrence of coronary artery aneurysms (CAAs) based on the accurate diagnosis and precise treatment of KD. Therefore, establishing a prediction model to identify KD from other febrile infectious diseases is crucial. In this report, we reviewed the clinical data of 216 patients with KD and 394 patients with other febrile infectious diseases and established a new prediction model with a high accuracy.

Peripheral blood total WBC is one of the predictors in this model and it increases in the acute phase of KD. WBCs can be used as a non-specific inflammatory indicator in combination with clinical manifestations to predict KD [20]. WBCs may also be able to predict the severity of systemic inflammation and IVIG non-reactivity in KD patients [21]. Other studies have shown that a WBC count greater than 16*10^9/L is positively correlated with heart damage [22]. Therefore, although the specificity of WBCs is not high for KD, it is widely used in clinical practice and has a practical significance for the clinical diagnosis of KD [23].

In this model, NLR (the ratio of the neutrophil count to the lymphocyte count of the peripheral blood) is an important predictor for identifying KD. The immune response to inflammation includes neutrophils moving to the site of inflammation, releasing inflammatory cytokines, and activating T cells, which play a key role in the development of vascular inflammation. Lymphocytes are produced by lymphoid organs and play an important role in the body’s immune response; they can also be used as a marker of immune regulation. Therefore, NLR is a reflection between the inflammatory response and the immunity balance. A few studies have shown that the higher the NLR value, the heavier the inflammatory response [24,25]. Recent studies have indicated that a high level of NLR is an independent influencing factor of IVIG resistance in KD [26].

Onodera’s PNI is an index reflecting the nutritional status. The PNI has been reported to be a strong indicator for predicting the prognosis of patients with malignant tumors and has been widely used in predicting the prognosis, postoperative complications, and quality of life of a variety of tumors [27,28]. The PNI was found to be a novel surrogate independent predictor for IVIG-resistant KD according to a recent study [29]. In KD patients, the ALB levels were significantly lower than those of the febrile control group and even lower in KD with CAL formation [30].

Recent studies have reported that a reduced lymphocyte count can serve as an independent predictor for IVIG resistance in KD [31]. Onodera’s PNI score is calculated based on these two indicators of lymphocytes and albumin and can reflect the nutritional status and immune function. In this study, Onodera’s PNI was an important predictor for distinguishing KD from other febrile diseases. 

The plasma level of C3 in the KD group was significantly higher than in the febrile controls and was also one of the important indicators for distinguishing KD from the controls. C3 is involved in the three complement pathways (classical, lectin, and alternative) and plays an important role in the innate immune response. Yan et al. found that, compared with a fever control group, the level of C3 was significantly higher in a KD group and it was higher in an IVIG-sensitive group compared with an IVIG-non-responsive group [7]. Dysregulation or overactivation of the complement system is the pathogenesis of vascular inflammation and aortic aneurysm formation [32,33]. However, few studies have addressed the complement pathway of KD [34,35]. Katayama et al. reported that a Ficolin 1 inhibitory antibody injection improved vasculitis of a KD mouse model, further suggesting that the lectin pathway may be involved in the pathogenesis of KD [36]. 

This study is a single-center retrospective study with a relatively small number of cases and a randomized controlled study with a larger sample of multiple centers is needed to further verify the value of the prediction model.

## 5. Conclusions

Our study demonstrated that a nomogram has a good prediction ability with WBC, NLR, C3, PNI, and other predictors. This report is the first to use C3 and the PNI as predictive factors to distinguish KD from febrile disease. This paper clarified the importance of C3 in KD and provided direction for further research on the pathogenesis of KD.

## Figures and Tables

**Figure 1 children-08-00825-f001:**
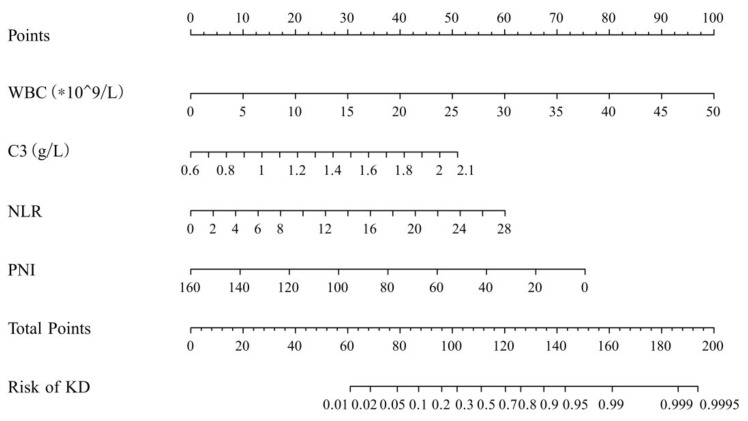
The nomogram for the probability of Kawasaki disease. WBC: white blood cell; C3: complement 3; NLR: neutrophil-to-lymphocyte ratio; PNI: prognostic nutritional index. KD: Kawasaki disease.

**Figure 2 children-08-00825-f002:**
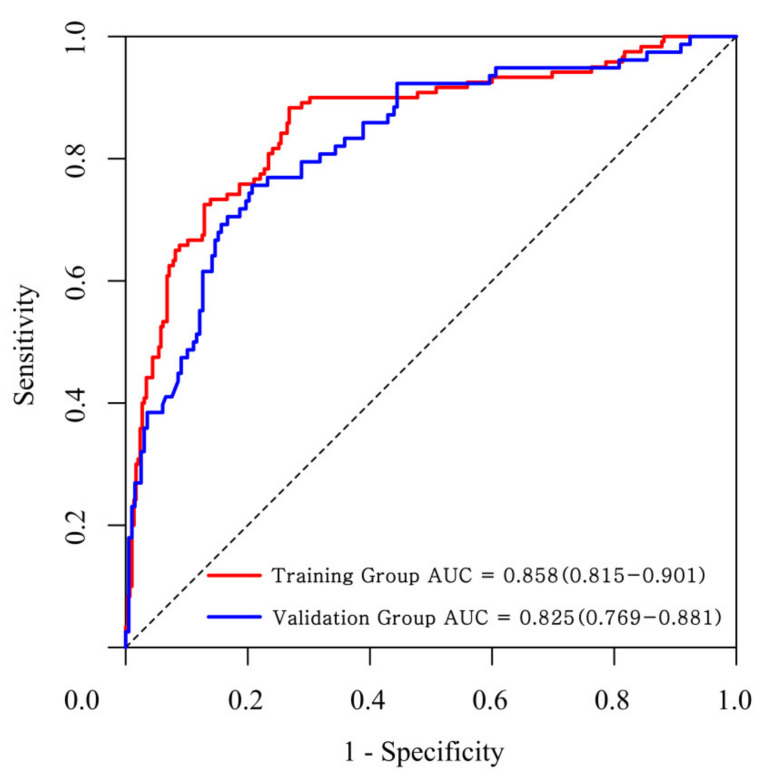
Receiver operating characteristic (ROC) curve of the nomogram for the training group and the validation group.

**Figure 3 children-08-00825-f003:**
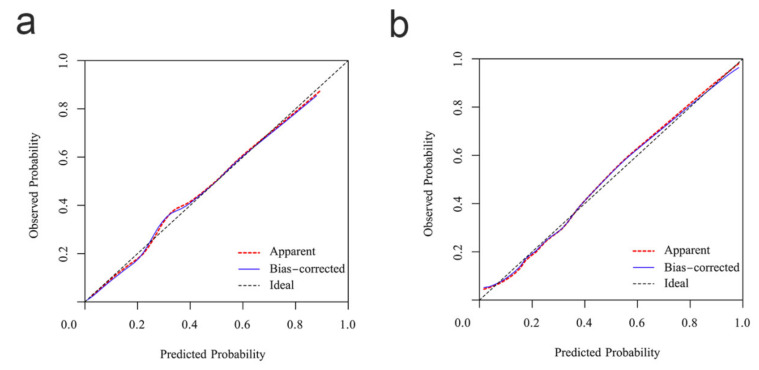
The calibration curves for the nomogram of the training group and the validation group. The black dashed line is the reference line for where an ideal nomogram would lie. The red dotted line is the performance of the nomogram and the blue solid line corrects for any bias in the nomogram. (**a**) The training group; (**b**) the validation group.

**Figure 4 children-08-00825-f004:**
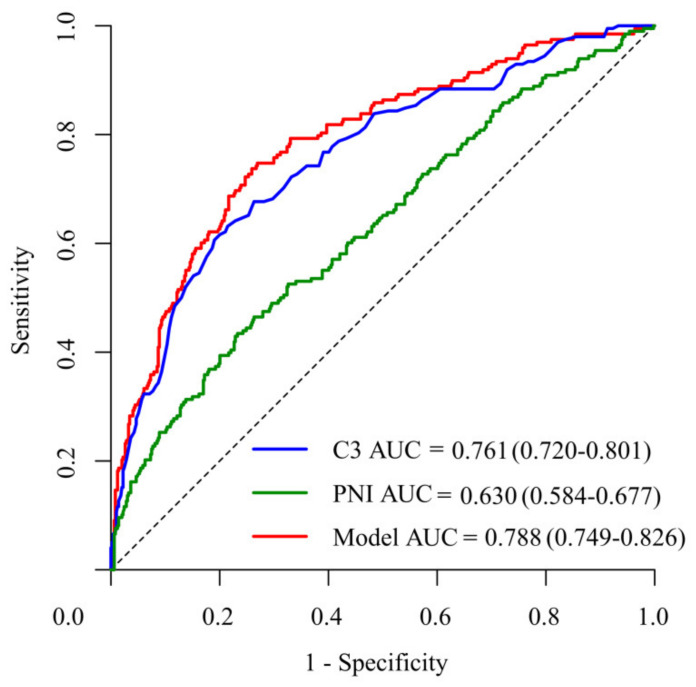
Receiver operating characteristic (ROC) curve of the C3 and the PNI and the ROC curve of the two variables.

**Table 1 children-08-00825-t001:** Comparison of the variables between the training group and the verification group.

Variable	Total (N = 692)	Training Group (N = 415)	Verification Group (N = 277)	*p*-Value
Age (month)	21 (11–40)	22 (12–40)	19 (11–40)	0.366
Male gender, N (%)	422 (60.98)	248 (59.76)	174 (62.82)	0.419
Body weight (Kg)	12.48 ± 4.73	12.66 ± 4.88	12.21 ± 4.5	0.215
WBC (*10^9/L)	10.4 (7.22–14.85)	10.2 (7.4–14.96)	11 (7.05–14.74)	0.955
IgA (g/L)	0.59 (0.4–1.01)	0.61 (0.41–1.03)	0.57 (0.37–0.96)	0.291
IgG (g/L)	7.27 ± 2.31	7.39 ± 2.33	7.1 ± 2.28	0.107
IgM (g/L)	1.04 (0.79–1.32)	1.04 (0.79–1.32)	1.03 (0.77–1.32)	0.766
C3 (g/L)	1.29 ± 0.27	1.28 ± 0.25	1.29 ± 0.28	0.532
C4 (g/L)	0.4 ± 0.13	0.4 ± 0.13	0.41 ± 0.14	0.401
NLR	1.3 (0.68–2.44)	1.31 (0.68–2.54)	1.27 (0.68–2.31)	0.371
PLR	83.85 (58.01–124.29)	83.07 (56.37–125.6)	84.98 (60.59–123)	0.576
PNI	60.92 ± 16.02	60.84 ± 15.28	61.05 ± 17.09	0.866

WBC: white blood cell; IgA: immunoglobulin A; IgG: immunoglobulin G; IgM: immunoglobulin M; C3: complement 3; C4: complement 4; NLR: neutrophil-to-lymphocyte ratio; PLR: platelet-to-lymphocyte ratio; PNI: prognostic nutritional index.

**Table 2 children-08-00825-t002:** Comparison between the Kawasaki disease patients and the febrile controls (the training group).

	Febrile Controls (N = 295)	KD (N = 120)	*p*-Value
Age (month)	22 (11–41)	22 (15–37)	0.537
Male gender, N (%)	178 (60.34)	70 (58.33)	0.706
Body weight (Kg)	12.87 ± 5.39	12.16 ± 3.27	0.104
WBC (*10^9/L)	9.1 (6.8–12.7)	14.36 (10.52–17.94)	<0.001 *
IgA (g/L)	0.59 (0.4–1.03)	0.64 (0.44–1.05)	0.473
IgG (g/L)	7.43 ± 2.28	7.27 ± 2.46	0.530
IgM (g/L)	1.04 (0.79–1.31)	1.05 (0.79–1.32)	0.895
C3 (g/L)	1.21 ± 0.23	1.44 ± 0.25	<0.001 *
C4 (g/L)	0.4 ± 0.12	0.39 ± 0.13	0.343
NLR	1 (0.57–1.89)	2.7 (1.55–4.69)	<0.001 *
PLR	76.86 (52.18–115.23)	99.49 (70.25–162.26)	<0.001 *
PNI	62.41 ± 14.8	56.97 ± 15.81	<0.001 *

KD: Kawasaki disease; WBC: white blood cell; IgA: immunoglobulin A; IgG: immunoglobulin G; IgM: immunoglobulin M; C3: complement 3; C4: complement 4; NLR: neutrophil-to-lymphocyte ratio; PLR: platelet-to-lymphocyte ratio; PNI: prognostic nutritional index; * *p* < 0.05.

**Table 3 children-08-00825-t003:** Multivariate logistic regression analysis.

	*p*-Value	OR	95% CI for OR	
			Lower	Upper
WBC (*10^9/L)	< 0.0001	1.201	1.121	1.288
C3 (g/L)	< 0.0001	22.631	6.867	74.585
NLR	0.025	1.218	1.025	1.446
PNI	0.002	0.958	0.932	0.984
Constant	< 0.0001	0.006		

WBC: white blood cell; C3: complement 3; NLR: neutrophil-to-lymphocyte ratio; PNI: prognostic nutritional index; OR: odds ratio; CI: confidence interval.

**Table 4 children-08-00825-t004:** The incidence risk of Kawasaki disease corresponding with the total score.

Total Points	Risk of KD
61	0.01
69	0.02
79	0.05
87	0.1
96	0.2
102	0.3
107	0.4
111	0.5
115	0.6
120	0.7
126	0.8
135	0.9
143	0.95
161	0.99
186	0.999
194	0.9995

**Table 5 children-08-00825-t005:** The cutoff of the C3 and the PNI.

	Cutoff	Sensitivity	Specificity	Yoden Index
C3 (g/L)	1.38	0.631	0.785	0.416
PNI	52.03	0.434	0.767	0.202

## Data Availability

The dataset containing the results from this article is available from the corresponding author upon request.

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
