# Peer review of "Complement 3 and the Prognostic Nutritional Index Distinguish Kawasaki Disease from Other Fever Illness with a Nomogram"

_children, 2021, doi:10.3390/children8090825_

Round 1

Reviewer 1 Report

The authors need to explain what the meaning of the  'training group' is.

Is there any comment about how to decide the line between incomplete KD and complete KD, or IVIG resistant KD and effective KD?

Author Response

Point 1 The authors need to explain what the meaning of the 'training group' is

Response 1: The training group was selected 60% randomly from the entire sample studied. (page2line89)

Point 2 Is there any comment about how to decide the line between incomplete KD and complete KD, or IVIG resistant KD and effective KD?

Response 2:IVIG responsiveness was defined as the abatement of fever within 48 hours after completing IVIG treatment and no return of fever (>38°C) for at least 7 days after, with marked improvement or normalization of the comorbid signs of inflammation. Incomplete KD was defined as those who had less than 4 symptoms and were finally diagnosed as KD.(Page2line80-84)

Reviewer 2 Report

The authors examined multiple factors to distinguish Kawasaki disease (KD) from other fever illnesses and found that increased C3 and decreased prognostic nutritional index (PNI) are likely to be present in KD. This is well conducted study and reasonably well written paper. I have a few minor suggestions.

Line 59. The authors state that the index was first proposed by Buzby and later established by Onodera. They reference 12 for Onodera, but this paper is by Wei and associates. I would suggest the original papers by Buzby and Onodera should be referenced and the remaining references re-numbered accordingly.

While there is a statistically significant difference in C3 and PNI between KD and other febrile illnesses, there appears to be considerable overlap. I would recommend that both these values be shown in a graph so that the reader can see the overlap. The authors should also indicate the cut-off values.

Author Response

Point1 Line 59. The authors state that the index was first proposed by Buzby and later established by Onodera. They reference 12 for Onodera, but this paper is by Wei and associates. I would suggest the original papers by Buzby and Onodera should be referenced and the remaining references re-numbered accordingly.

Response 1We have referenced the original paper by Buzby and Onodera. (page2line59)

Point 2 While there is a statistically significant difference in C3 and PNI between KD and other febrile illnesses, there appears to be considerable overlap. I would recommend that both these values be shown in a graph so that the reader can see the overlap. The authors should also indicate the cut-off values.

Response 2: The area under the curve of ROC of C3 is 0.761(95% confidence interval: 0.72-0.801), 0.63 (95% confidence interval: 0.584-0.677) in PNI, and 0.788(95% confidence interval:0.749-0.826) in the two variables combine. ( Figure 4). The cutoff of C3 and PNI was 1.38g/L and 52.03. ( Table5)(page6line163-166 and page7line167-171)
